# A Reconfigurable Polarimetric Photodetector Based on the MoS_2_/PdSe_2_ Heterostructure with a Charge-Trap Gate Stack

**DOI:** 10.3390/nano14231936

**Published:** 2024-12-01

**Authors:** Xin Huang, Qinghu Bai, Yang Guo, Qijie Liang, Tengzhang Liu, Wugang Liao, Aizi Jin, Baogang Quan, Haifang Yang, Baoli Liu, Changzhi Gu

**Affiliations:** 1Beijing National Laboratory for Condensed Matter Physics, Institute of Physics, Chinese Academy of Sciences, Beijing 100190, China; baiqinghu@iphy.ac.cn (Q.B.); yangguo@aphy.iphy.ac.cn (Y.G.); azjin@iphy.ac.cn (A.J.); quanbaogang@iphy.ac.cn (B.Q.); hfyang@iphy.ac.cn (H.Y.); blliu@iphy.ac.cn (B.L.); 2School of Physical Sciences, CAS Key Laboratory of Vacuum Physics, University of Chinese Academy of Sciences, Beijing 100190, China; 3Songshan Lake Materials Laboratory, Dongguan 523808, China; liangqijie@sslab.org.cn; 4State Key Laboratory of Radio Frequency Heterogeneous Integration, Shenzhen University, Shenzhen 518060, China; 2200434018@email.szu.edu.cn (T.L.); wgliao@szu.edu.cn (W.L.); 5College of Electronics and Information Engineering, Shenzhen University, Shenzhen 518060, China; 6CAS Center for Excellence in Topological Quantum Computation, CAS Key Laboratory of Vacuum Physics, University of Chinese Academy of Sciences, Beijing 100190, China

**Keywords:** polarimetric photodetector, reconfiguration, two-dimensional heterostructure, charge-trap gate stack

## Abstract

Besides the intensity and wavelength, the ability to analyze the optical polarization of detected light can provide a new degree of freedom for numerous applications, such as object recognition, biomedical applications, environmental monitoring, and remote sensing imaging. However, conventional filter-integrated polarimetric sensing systems require complex optical components and a complicated fabrication process, severely limiting their on-chip miniaturization and functionalities. Herein, the reconfigurable polarimetric photodetection with photovoltaic mode is developed based on a few-layer MoS_2_/PdSe_2_ heterostructure channel and a charge-trap structure composed of Al_2_O_3_/HfO_2_/Al_2_O_3_ (AHA)-stacked dielectrics. Because of the remarkable charge-trapping ability of carriers in the AHA stack, the MoS_2_/PdSe_2_ channel exhibits a high program/erase current ratio of 10^5^ and a memory window exceeding 20 V. Moreover, the photovoltaic mode of the MoS_2_/PdSe_2_ Schottky diode can be operated and manipulable, resulting in high and distinct responsivities in the visible broadband. Interestingly, the linear polarization of the device can be modulated under program/erase states, enabling the reconfigurable capability of linearly polarized photodetection. This study demonstrates a new prototype heterostructure-based photodetector with the capability of both tunable responsivity and linear polarization, demonstrating great potential application toward reconfigurable photosensing and polarization-resolved imaging applications.

## 1. Introduction

The technology of polarization-sensitive photodetection plays a vital role in both civilian and military fields, such as biomedical imaging, quantum communication, and three-dimensional (3D) holographic displays [1,2,3]. In the past, conventional polarimetric photodetectors required the integration of a prepositive polarizer, lens, or polarization coding system, which increased the fabrication complexity and cost of the imaging systems [4]. Therefore, developing the architecture of polarizer-free polarimetric photodetection becomes crucial for satisfying the needs of on-chip integration, miniaturization, and multi-functionalities.

Recently, low symmetric two-dimensional (2D) semiconductors with in-plane anisotropic crystal structures have shown great potential for linear polarization photodetection. This capability arises from their intrinsic linear dichroism as well as the absence of surface dangling bonds, which allows for integration into complex heterostructures regardless of lattice mismatch [5,6,7,8]. Among 2D materials and their heterojunctions, including black phosphorous [9], ReS_2_ [10], GeAs [11], 1T′-MoTe_2_ [12], 1T′-WTe_2_ [13], as well as WSe2/ReSe_2_ [14], WS_2_/GeAs [15], and graphene/PdSe_2_/Ge [2], they have been widely used to build polarization-sensitive photodetectors with high detectivity, fast speed, and broad-band sensitivity.

Some previous architectures of the device have been proposed; however, the existing technologies have still been unable to realize multifunctional photodetection with tunable responsivity and polarization sensitivity. The capability for reconfigurable polarimetric photodetection could enable higher-resolution polarimetric imaging. To achieve tunable responsivity and polarization sensitivity, configurations such as a split-gate configuration [16] or ferroelectric polarization [17] have been employed. However, these approaches suffer from the complicated design of four electrical terminals and the high energy consumption required. At the same time, nonvolatile polarimetric photodetection, in which the reconfigurable responsivity and polarization sensitivity are necessary schemes for the integration of multi-functional modules, realizes an “All-in-one” system, such as in-memory sensing technology [18], vision acquisition [19], and high-level cognitive computing [20]. However, the tunable nonvolatile and reconfigurable polarimetric photodetection in 2D devices still remains rarely studied.

This work demonstrates a novel multifunctional photodetector engineered to offer reconfiguration in both responsivity and polarization sensitivity. It is developed on a few-layer MoS_2_/PdSe_2_ heterostructure and an Al_2_O_3_/HfO_2_/Al_2_O_3_ (AHA) charge-trap gate stack. Our elaborately designed photodetector exhibits a remarkable photovoltaic photodetection performance under visible illumination, which can be attributed to the built-in electrical driving effect of the MoS_2_/PdSe_2_-based Schottky diode. Under modulation of the AHA charge-trap gate stack, the electrical characteristics of the MoS_2_/PdSe_2_ Schottky diode can be tuned and maintained at the program/erase state, exhibiting an unprecedented memory window exceeding 20 V and the program/erase current ratio of 10^5^. Moreover, the photovoltaic mode of the MoS_2_/PdSe_2_ Schottky diode is operated and switchable, resulting in high and distinct responsivities in the visible spectral band. Interestingly, linear polarization can be further modulated under the program/erase state, enabling the reconfigurable capability of linearly polarized photodetection. Our work provides promising solutions for increasing the versatility of applications for reconfigurable photodetection. Importantly, the charge-trap gate stack was first applied on the 2D heterostructure to engineer the band alignment type, enhance the photodetection performance, and enrich functionalities.

## 2. Materials and Methods

MoS_2_ is one of the most studied 2D materials, and it demonstrates the ability to have remarkable electronic and optoelectronic properties, which makes it a great potential photodetector candidate. Considering the large carrier density and high work function of PdSe_2_, a depletion region of the junction can be formed by stacking the MoS_2_/PdSe_2_ heterostructure. Multilayers of MoS_2_ and PdSe_2_ were subsequently exfoliated and stacked together (Figure 1a). The details of device fabrication are provided in Appendix A. The schematic of the MoS_2_/PdSe_2_ photodetector is shown in Figure 1b. Figure 1c shows the distinct Raman peaks of the MoS_2_/PdSe_2_ heterostructure, which correspond to 382 cm^−1^ ($E_{2g}^{1}$) and 407 cm^−1^ ($A_{1g}$) for MoS_2_ and 144 cm^−1^ ($A_{g}^{1}$), 203 cm^−1^ ($A_{g}^{2}$), 222 cm^−1^ ($B_{1g}^{2}$), and 258 cm^−1^ ($A_{g}^{3}$) for PdSe_2_. Figure 1d and e show the I_ds_–V_ds_ curves under different gate voltages V_G_ for MoS_2_ and PdSe_2_ field-effect transistor (FET), respectively. Notably, both I_ds_–V_ds_ curves demonstrate obvious linearity, and it can be proved to be Ohmic source-drain contact for both FETs, which is necessary for optoelectronic characteristics of the MoS_2_/PdSe_2_ photodetector. Furthermore, the output curves of I_ds_–V_ds_ for MoS_2_ FET exhibit a n-type ambipolar conducting behavior at V_G_ from −5 to 5 V. By contrast, the current slightly increases with decreasing negative V_G_ for PdSe_2_ FET, which indicates the semi-metallic behavior. Given the bandgaps of multilayer MoS_2_ and PdSe_2_ are previously reported to be 1.2 and 0.03 eV [21,22], the band alignment of the MoS_2_/PdSe_2_ heterojunction is illustrated in Figure 1f. The Schottky barrier can be formed at the interface of the MoS_2_/PdSe_2_ heterojunction, and the I_ds_–V_ds_ curve exhibits a rectification behavior, demonstrating a rectification ratio of I_on_/I_off_ up to 10.

## 3. Results

### 3.1. Transfer Characteristics and Static Memory Behavior

A charge-trap stack of Al_2_O_3_/HfO_2_/Al_2_O_3_ (6 nm/8 nm/32 nm) was deposited via atomic layer deposition (ALD). Figure 2a shows the transfer curves of the MoS_2_/PdSe_2_ Schottky diode. The transfer curves were acquired by sweeping the gate voltage V_G_ in a closed loop (from negative to positive values) under a fixed V_ds_ of −1 V, exhibiting a clear hysteresis window, and the hysteresis enables widening as the V_G_ sweep range increases from 5 to 25 V. The I_ds_–V_G_ curves exhibit a clockwise memory window. The extraction of memory window ∆V increases almost linearly with the maximum V_G_ and reaches a maximum of 20 V when the V_G_ sweeps to 25 V (Figure 2b). The transfer curve of I_ds_–V_G_ decreases with increasing negative V_G_, suggesting that n-type MoS_2_ dominates the transfer characteristics of the device. Figure 2c illuminates the device operation process. When a high positive/negative V_G_ was applied to the gate, the band alignment started favoring the tunneling in/out of electrons from the MoS_2_/PdSe_2_ channel to the HfO_2_ charge-trap layer, which resulted in the change of carrier concentration in MoS_2_ and a switch between program and erase state, respectively.

### 3.2. Dynamic Memory Behavior of the Device

The transfer characteristics of the device are further studied under different biases. As shown in Figure 3a,b, they show that an obvious memory window occurs under both a forward bias of −1 V and a reverse bias of +1 V. A maximum program/erase current ratio of 10^5^ can be achieved under a forward bias of −1 V. Endurance and retention time of the MoS_2_/PdSe_2_ memory are provided in Appendix A. To explore the dynamic transition of the device, the device was initially set into an erase state by applying a negative gate pulse (−10 V, duration of 2 s). The output curves I_ds_–V_ds_ were read by sweeping V_ds_ from −1 V to +1 V after applying a series of +25 V gate pulses with different duration times. The output curve I_ds_–V_ds_ shows a clear decrease and is nearly saturated when the width of the pulse increases to 0.2 s (Figure 3c). According to the expression of the charge-trapping rate [23], the calculated charge-trapping rate varies from 10^15^ cm^−2^s^−1^ to 10^14^ cm^−2^s^−1^ when the pulse width changes from 0.01 s to 0.2 s. Figure 3d shows the dependency of output curves I_ds_–V_ds_ with the amplitude of the gate pulse. It demonstrates that output current decreases with the increase of pulse amplitude. This can be explained by the modulation of the Schottky barrier through the gate pulse, which also suggests that the charge-trapping mechanism of the AHA gate stack dominates the memory behavior of the device.

### 3.3. Photovoltaic Behavior and Reconfigurable Linear Polarization

Given the excellent memory switching properties of the device (including unprecedented memory window, large program/erase current ratio, and nonvolatile switchable Schottky barriers) and strong optical anisotropy of PdSe_2_, the polarization-modulated photovoltaic behavior in the MoS_2_/PdSe_2_-based photodetector was worthy of investigation. To characterize it, a positive (+25 V) and negative (−10 V) gate pulse with a width of 0.2 s were applied to switch the device into the program and erase state, respectively. In the program/erase state, I_ds_–V_ds_ characteristics under illumination were recorded by using the polarized 520 nm light with an intensity of 120 mW/cm^2^. Figure 4a,b show the I_ds_–V_ds_ characteristics of the device under parallel ($0^{{}^{\circ}}$) and vertical ($90^{{}^{\circ}}$) polarized light in the program and erase state, respectively ($0^{{}^{\circ}}$ and $90^{{}^{\circ}}$ directions correspond to the b-axis and a-axis crystalline direction of PdSe_2_). The crystalline orientation of PdSe_2_ was determined by angle-resolved polarized Raman spectroscopy (ARPRS), and details of the measurement are provided in Appendix A. It can be noticed that the device exhibits noticeable photovoltaic responses, including a short-circuit current (I_sc_) of ~15 nA/30 nA and an open-circuit voltage (V_oc_) of ~−0.014 V/−0.012 V in the program/erase state under parallel light excitation. After switching the polarization of light to the vertical direction, the illuminated I_ds_–V_ds_ curves shift toward the higher value, showing that I_sc_ and V_oc_ increase to ~54 nA/82 nA and ~−0.027/−0.025 V in the program/erase state. It is noted that significant photocurrent noise occurs during photocurrent measurement, which can be attributed to the degeneration of 2D flakes and environmental noise. The protection layer of Al_2_O_3_ on top of the MoS_2_/PdSe_2_ heterostructure and the optimized measurement method can significantly reduce the noise, which is summarized in Appendix A. In addition, gate voltage was applied to modulate the performance of the device. The responsivity was extracted and plotted in Figure 4c at different memory states and polarization of light. The noise equivalent power (NEP) and detectivity of this photodetector were measured and are provided in Appendix A. The device shows typical transfer characteristics of an n-type MoS_2_ semiconductor. As the gate voltage increases from −2 V to 2 V, all of the responsivities increase at V_ds_ = −1 V. A figure of merit of the linear polarized photodetection is the degree of linear polarization (LP), where LP = (I_max_ − I_min_)/(I_max_ + I_min_), where I_max_ and I_min_ are the photocurrents of the detected light parallel and perpendicular to the primary polarization direction, respectively. Figure 4d shows the LP results as a function of V_G_. Under 120 mW/cm^2^ light illumination, both LP values of the device gradually increase from 0.4/0.36 to 0.55/0.49 in the program/erase state. This indicates that the LP of the device can be effectively modulated by the memory state, and its gap between the program and erase state becomes more obvious under the positive gate voltage.

## 4. Discussion

To understand the photoresponse mechanism of the MoS_2_/PdSe_2_ device, the energy band structure diagram is illustrated in Figure 5. Since the fermi level of PdSe_2_ is lower than MoS_2_, the electrons will flow from MoS_2_ to PdSe_2_, and the holes diffuse in opposite directions to MoS_2_. The opposite diffusion of electrons and holes introduces a Schottky barrier with the build-in field **E_in_** pointing from MoS_2_ to PdSe_2_, which is described in Figure 1f. When the laser shines on the surface of the device, the electrons confined in the valence band will be excited by the conduction bands in both materials. Then, with the help of build-in field **E_in_**, the electrons occupied in PdSe_2_ can be driven to the conduction band of MoS_2_, while **E_in_** will force the holes within the MoS_2_ valence band to flow into the valence band of PdSe_2_, resulting in the photovoltaic behavior. When the negative gate voltage pulse is applied to switch the device into an erase state (Figure 5b), the energy band of MoS_2_ is lowered, and **E_in_** will increase, which results in the enhancement of separation of photo-generated electron-hole pairs as well as short-circuit current. Meanwhile, the photocurrent generated from MoS_2_ increases in the erase state, which results in the decrease of LP because of the intrinsic polarization-insensitivity of MoS_2_. On the other hand, the **E_in_** will be reduced in the program state (Figure 5c), which induces the decrease of short-circuit current and increase of LP. In this way, we can adjust the energy band structure of MoS_2_/PdSe_2_ by switching the program/erase state, thereby adjusting the photodetection performance of the device.

## 5. Conclusions

In summary, a multifunctional photovoltaic photodetector was demonstrated, which was composed of in-plane anisotropic PdSe_2_ and MoS_2_ with an AHA charge-trap gate stack. The device exhibits a nonvolatile phenomenon in both electrical and photovoltaic characteristics, resulting from the modulation of band alignment by the gate voltage pulse. Utilizing the AHA charge-trap gate stack, the memory window and program/erase current ratio of MoS_2_/PdSe_2_ can be effectively modulated. Acting as a reconfigurable polarimetric photodetector, the device exhibits a reversible performance of both responsivity and polarization-sensitive photocurrent by switching the program and erase state, rendering it a promising candidate for polarization signal recognition and imaging.

## Figures and Tables

**Figure 1 nanomaterials-14-01936-f001:**
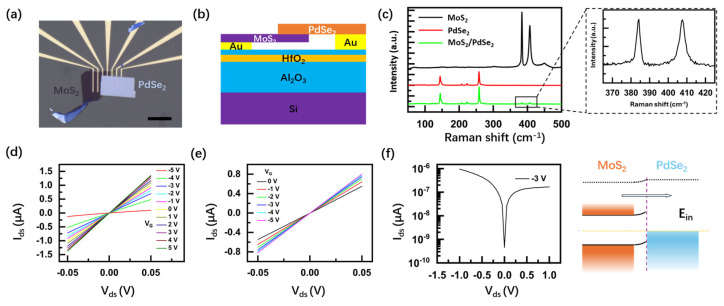
Schematics and characterization of the MoS_2_/PdSe_2_ heterostructure device: (**a**,**b**) Schematic and picture of MoS_2_/PdSe_2_ heterostructure photodetector. The scale bar is 20 μm; (**c**) Raman spectra of the multilayer MoS_2_ flakes, PdSe_2_ flakes, and heterostructure, respectively; (**d**,**e**) I_ds_–V_ds_ relationship of MoS_2_ and PdSe_2_, respectively; (**f**) the transfer characteristics of the MoS_2_/PdSe_2_ heterostructure under the bias of −3 V and schematic of the MoS_2_/PdSe_2_ heterostructure-based Schottky barrier.

**Figure 2 nanomaterials-14-01936-f002:**
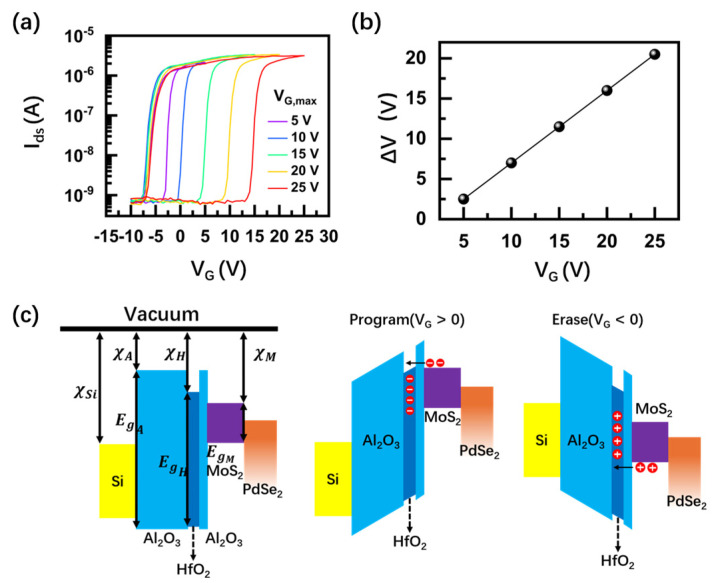
The statical behavior of the nonvolatile gate charge-trap memory based on MoS_2_/PdSe_2_ heterostructure: (**a**) I_ds_–V_G_ characteristics of the device under different V_G_ at V_ds_ = −1 V; (**b**) extraction of memory window ∆V vs. V_G_. The memory window increases from 1 to ∼20 V in our experimental settings; (**c**) band diagram of the program/erase state of the device under positive and negative V_G_. Positive V_G_ programs the device. Electrons tunneling from the few-layer MoS_2_ channel are accumulated in the HfO_2_ charge-trap layer. Negative V_G_ erases the device. Holes tunnel from the few-layer MoS_2_ channel to the HfO_2_ charge-trap layer.

**Figure 3 nanomaterials-14-01936-f003:**
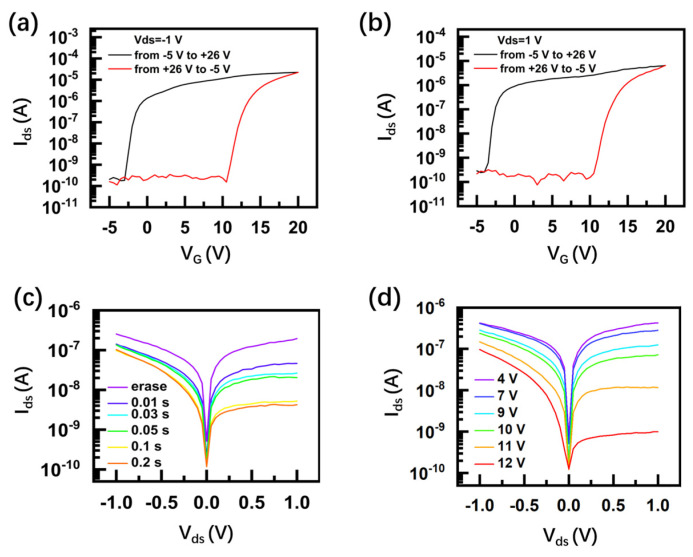
The dynamic behavior of the nonvolatile gate charge-trap memory based on MoS_2_/PdSe_2_ heterostructure: (**a**,**b**) I_ds_–V_G_ characteristics of the device under different V_G_ under the forward bias of −1 V and reverse bias of +1 V, respectively; (**c**,**d**) I_ds_–V_ds_ characteristics of the device under different pulse durations and amplitudes.

**Figure 4 nanomaterials-14-01936-f004:**
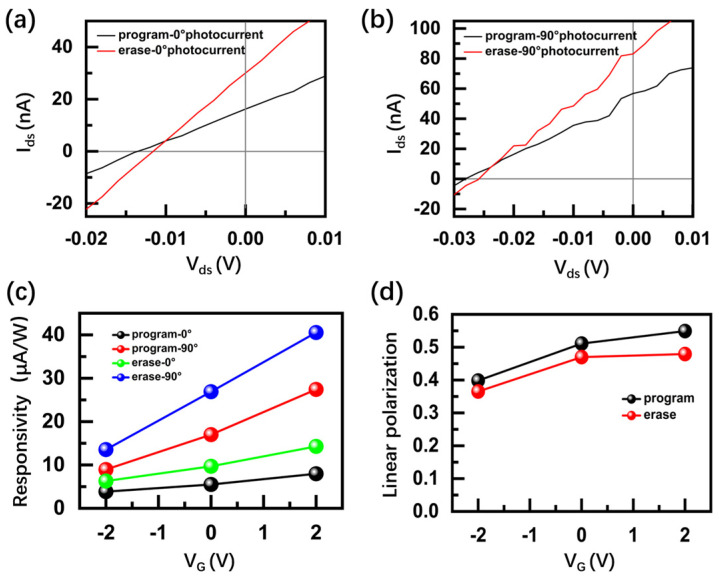
The polarization-modulated photovoltaic behavior of MoS_2_/PdSe_2_ photodetector: (**a**,**b**) Short-circuit current I_sc_ and open-circuit voltage V_oc_ of MoS_2_/PdSe_2_ photodetector under the program and erase state, respectively. (**c**,**d**) Dependency of responsibility and linear polarization with gate voltage V_G_ under different program/erase states and polarization directions of light.

**Figure 5 nanomaterials-14-01936-f005:**
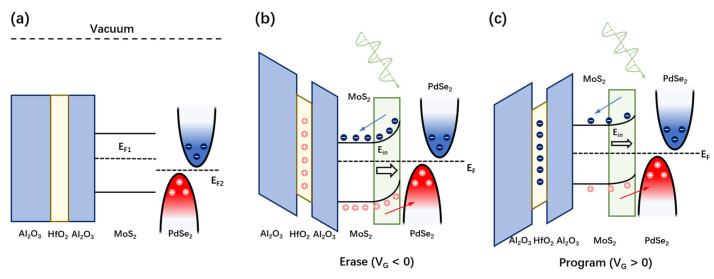
Photoresponse mechanism of reconfigurable MoS_2_/PdSe_2_ photodetector: (**a**) The energy band structure of AHA charge-trap stack and MoS_2_/PdSe_2_ heterostructure before contact; (**b**,**c**) the energy band structure of the device and the flow of photo-generated electron-holes under illumination when the device is set to erase (V_G_ < 0) and program states (V_G_ > 0), respectively.

## Data Availability

The data presented in this study are available on request from the corresponding author.

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
