# Peer review of "A Reconfigurable Polarimetric Photodetector Based on the MoS2/PdSe2 Heterostructure with a Charge-Trap Gate Stack"

_nanomaterials, 2024, doi:10.3390/nano14231936_

Round 1

Reviewer 1 Report

Comments and Suggestions for Authors

In this manuscript, Xin Huang et al. present a research study on a reconfigurable polarimetric photosensor based on a MoSâ‚‚/PdSeâ‚‚ Schottky junction. The manuscript is both timely and interesting. However, before publication in Nanomaterials, the following questions and remarks should be addressed in the manuscript to ensure clarity and completeness of the study:

  1. Why were MoSâ‚‚ and PdSeâ‚‚ used as few- or multi-layer flakes instead of single-layer flakes (as most of researchers are looking for)? What are the thicknesses or number of layers for these materials? The authors should provide such information using the Raman spectra measurements presented in the paper or via additional measurements such as profilometry/AFM techniques.

  2. The exfoliation and stacking techniques used in the fabrication process must be described in more detail. Which specific methods were employed? Was a PDMS-based exfoliation technique used here? Providing this information would enhance the reproducibility of the results.

  3. The authors should provice carrier mobilities for MoSâ‚‚, PdSeâ‚‚, and the MoSâ‚‚/PdSeâ‚‚ FETs.

  4. Have the authors considered using graphene as semimetallic layer instead of PdSeâ‚‚? Could graphene improve the performance of such devices? Maybe including a discussion/perspectives on the potential advantages or disadvantages of using PdSe2 or graphene in this context would be valuable.

  5. In terms of photodetection experiments: how did the authors identify the crystallographic directions (b-axis and a-axis) of PdSeâ‚‚?

  6. How was the responsivity calculated? Was it simply determined as photoresponse/power, or did the calculation account for device dimensions and spot areas?

  7. What is the physical explanation for the observed differences in photocurrent when varying the light polarization from parallel to perpendicular? Did the authors investigate the response to circularly polarized light as well?

  8. English should be improved.

Comments on the Quality of English Language

English throughout the manuscript needs improvement. Some sentences are too long and overly complex, awkward or grammatically incorrect. Please check it carefuly.

Reviewer 2 Report

Comments and Suggestions for Authors

The authors submitted a report dealing with a MoS2/PdSe2 heterostructure with Al2O3/HfO2/Al2O3 dielectrick stack serving as a charge-trapping centers. The proposed system works as a Schottky diode with very poor rectification behaviour, but really excellent memory window (because of the electrostatic charge stored in the dielectric stack). Up to this point it is not novel at all; however, the authors report also the sensitivity to the incident beam polarisation. Nevertheless, there are sebveral issues to fix prior the discussion about the publicatikon acceptance.

(1) Memory device properties
If authors illustrate the memory device properties, they should evaluate also the retention time.

(2) Noise discussion
Interestingly, the current due to the 90° polarisation is recorded with significant noise even though the 0° polarisation does not have such noise (even though it is lower current). This should be discussed and explained. Is it because of the device instability?

(3) Photodevice properties
Since authors illustrate the device as a photodetector, the paratmeters such as photosensitivity, noise equivalent power, and detectivity (in Jones) should be evaluated.

(4) Proofreading
The manuscript contains various typos (e.g. trapping rate is "cm^-2 t^-1" instead of "cm^-2 s^-1"), please proofread again to avoid them.

As a result, I am willing to support the submitted manuscript if the above-mentioned issues will be solved.

Round 2

Reviewer 1 Report

Comments and Suggestions for Authors

The authors have addressed all my previous remarks. The manuscript has been improved and can be published in Nanomaterials.

Reviewer 2 Report

Comments and Suggestions for Authors

The authors responded to all questions and modified the manuscript according to all suggestions. As a result, I have no objections and recommend accepting the submitted manuscript.